# WEAKLY SUPERVISED CAUSAL REPRESENTATION LEARNING

**Johann Brehmer,**[*, 1] **Pim de Haan,**[*, 1 2] **Phillip Lippe,**[2] **and Taco Cohen**[1]
[1]Qualcomm AI Research[†]        [2]QUVA Lab, University of Amsterdam
{jbrehmer, pim, tacos}@qti.qualcomm.com; p.lippe@uva.nl

## ABSTRACT

Learning high-level causal representations together with a causal model from unstructured low-level data such as pixels is impossible from observational data alone. We prove under mild assumptions that this representation is identifiable in a weakly supervised setting. This requires a dataset with paired samples before and after random, unknown interventions, but no further labels. Finally, we show that we can infer the representation and causal graph reliably in a simple synthetic domain using a variational autoencoder with a structural causal model as prior.

## 1 INTRODUCTION

The dynamics of many systems can be described in terms of some high-level variables and causal relations between them. Often, these causal variables are not known but only observed in some unstructured, low-level representation, such as the pixels of a camera feed. Learning the causal representations together with the causal structure between them is a challenging problem and important for instance for applications in robotics and autonomous driving (Schölkopf et al., 2021). Without prior assumptions on the data-generating process or supervision, it is impossible to identify the causal variables and their causal structure uniquely (Eberhardt, 2016; Locatello et al., 2019).

In this work, we show that a weak form of supervision is sufficient to identify both the causal representations and the structural causal model between them. We consider a setting in which we have access to data pairs, representing the system before and after a randomly chosen unknown intervention. Neither labels on the intervention targets nor active control of the interventions are necessary for our identifiability theorem, making this setting useful for offline learning. We prove that with this form of weak supervision, latent causal models (LCMs)—

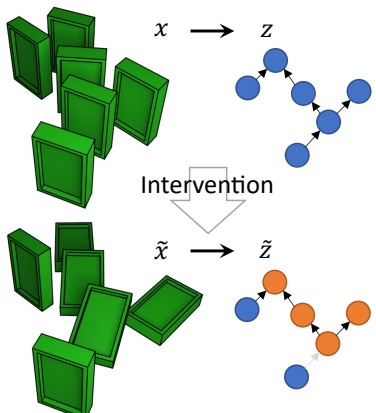

Figure 1: We learn to represent pixels $x$ as causal variables $z$. The bottom shows the effect of intervening on the orange variable. We prove that variables and causal model can be identified from samples $(x, \tilde{x})$.

structural causal models (SCMs) together with a decoder from the causal factors to the data space— are identifiable up to a relabelling and elementwise reparameterizations of the causal variables.

We then introduce a practical implementation for LCM inference by using an SCM as a prior in a variational autoencoder (VAE). In a range of experiments, we show that LCMs can learn the true causal variables and the causal structure from unstructured data.

**Related work**    Our work builds on the approach of Locatello et al. (2020) to *disentangled representation learning*. The authors introduce a similar weakly supervised setting where observations are collected before and after unknown interventions. In contrast to our work, however, they focus on disentangled representations, i. e. independent factors of variation with a trivial causal graph, which our work subsumes as a special case. Other relevant works on disentangled representation learning

---

[*]Equal contribution
[†]Qualcomm AI Research is an initiative of Qualcomm Technologies, Inc.

and (nonlinear) independent component analysis include Hyvärinen & Oja (2000); Shu et al. (2020); Khemakhem et al. (2020); Hälvä et al. (2021) and Lachapelle et al. (2022).

The problem of *causal representation learning* has been gaining attention lately, for a recent review see Schölkopf et al. (2021). Lu et al. (2021) learn causal representations by observing similar causal models in different environments. Von Kügelgen et al. (2021) use the weakly supervised setting to study self-supervised learning, using a known but non-trivial causal graph between content and style factors. Lippe et al. (2022b) learn causal representations from time-series data from labelled interventions, assuming that causal effects are not instantaneous but can be temporally resolved. Yang et al. (2021) propose to train a VAE with an SCM prior, but require the true causal variables as labels. To the best of our knowledge, our work is the first to provide identifiability guarantees for arbitrary, unknown causal graphs in this weakly supervised setting.

## 2 IDENTIFIABILITY OF LATENT CAUSAL MODELS FROM WEAK SUPERVISION

In this section, we show theoretically that causal variables and causal mechanisms are identifiable from weak supervision. In Sec. 3 we will then demonstrate how we can learn causal models in practice by training a causally structured VAE.

**Setup** We begin by defining latent causal models and the weakly supervised setting. Here, we only provide informal definitions and assume familiarity with common concepts from causality; see Appendix A.1 for a complete and precise treatment. In Appendix A.3, we discuss limitations of the setup and possible generalizations.

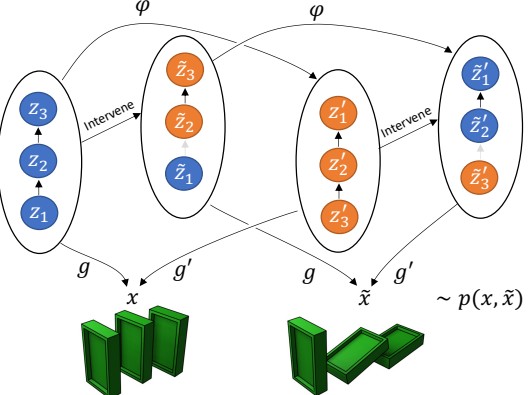

We describe the causal structure between latent variables as an Structural Causal Model (SCM). An SCM $\mathcal{C}$ describes the relation between causal variables $z_1, \ldots, z_n$ with domains $\mathcal{Z}_i$ and noise variables $\epsilon_1, \ldots, \epsilon_n$ with domains $\mathcal{E}_i$ along a directed acyclic graph $\mathcal{G}$. Causal mechanisms $f_i : \mathcal{E}_i \times \prod_{j \in \mathbf{pa}_i} \mathcal{Z}_j \to \mathcal{Z}_i$ describe how the value of a causal variable is determined from the associated noise variables as well as the values of its parents in the graph. Finally, an SCM includes a probability measure for the noise variables.

Figure 2: In LCM $\mathcal{M}$, $z_i$ denotes whether the $i$-th stone from the front is standing. Intervening on the second variable leads to $\tilde{z}$. The decoder $g$ renders $z, \tilde{z}$ as pixels $x, \tilde{x}$. LCM $\mathcal{M}'$ has an equivalent representation in which $z_i'$ denotes whether the $i$-th stone from the *back* has *fallen*. In Theorem 1, we prove that if and only if two causal models have the same pixel distribution $p(x, \tilde{x})$, there exists an LCM isomorphism $\varphi$: an element-wise reparametrization of the causal variables plus a permutation of the ordering that commutes with interventions and causal mechanisms.

An SCM entails a unique solution $s : \mathcal{E} \to \mathcal{Z}$ defined by successively applying the causal mechanisms. We require the structural equations to be pointwise diffeomorphic, $s$ is thus also diffeomorphic. It also entails an observational distribution $p_{\mathcal{C}}(z)$ (Markov with respect to the graph of the SCM), which is the pushforward of $p_{\mathcal{E}}$ through the solution.

A perfect intervention $(I, (\tilde{f}_i)_{i \in I})$ modifies an SCM by replacing for a subset of the causal variables, called the intervention target set $I \subset \{1, ..., n\}$, the causal mechanism $f_i$ with a new mechanism $\tilde{f}_i : \mathcal{E}_i \to \mathcal{Z}_i$, which does not depend on the parents. The intervened SCM has a new solution $\tilde{s}_I : \mathcal{E} \to \mathcal{Z}$. We define interventions to be atomic if the number of targeted variables is one or zero.

We will reason about generative models in a data space $\mathcal{X}$, in which the causal structure is latent. Also including a distribution of interventions, we define LCMs:

**Definition 1** (Latent causal model (LCM)). *A latent causal model $\mathcal{M} = \langle \mathcal{C}, \mathcal{X}, g, \mathcal{I}, p_{\mathcal{I}} \rangle$ consists of*

- *an acyclic SCM $\mathcal{C}$ that admits a faithful distribution,*
- *an observation space $\mathcal{X}$,*
- *a decoder $g : \mathcal{Z} \to \mathcal{X}$ that is diffeomorphic onto its image,*
- *a set $\mathcal{I}$ of interventions on $\mathcal{C}$, and*
- *a probability measure $p_{\mathcal{I}}$ over $\mathcal{I}$.*

We define two LCMs as equivalent if all of their components are equal up to a permutation of the causal variables and elementwise diffeomorphic reparameterizations of each variable, see Fig. 2.

**Definition 2** (LCM isomorphism (informal)). *Let $\mathcal{M} = \langle \mathcal{C}, \mathcal{X}, g, \mathcal{I}, p_{\mathcal{I}} \rangle$ and $\mathcal{M}' = \langle \mathcal{C}', \mathcal{X}, g', \mathcal{I}', p'_{\mathcal{I}'} \rangle$ be two LCMs with identical observation space. An LCM isomorphism between them is a graph isomorphism $\psi : \mathcal{G}(\mathcal{C}) \rightarrow \mathcal{G}(\mathcal{C}')$ together with elementwise diffeomorphisms for noise and causal variables that tell us how to reparameterize them, such that the structure functions, noise distributions, decoder, intervention set, and intervention distribution of $\mathcal{M}'$ are compatible with the corresponding elements of $\mathcal{M}$ reparameterized through the graph isomorphism and elementwise diffeomorphisms. $\mathcal{M}$ and $\mathcal{M}'$ are equivalent, $\mathcal{M} \sim \mathcal{M}'$, if and only if there is an LCM isomorphism between them.*

Following Locatello et al. (2020), we define a generative process of pre and post interventional data:[1]

**Definition 3** (Weakly supervised generative process). *Consider an LCM $\mathcal{M}$ where the underlying SCM has continuous noise spaces $\mathcal{E}_i$, independent probabilities $p_{\mathcal{E}_i}$, and admits a solution $s$. We define the weakly supervised generative process of data pairs $(x, \tilde{x}) \sim p_{\mathcal{M}}^{\mathcal{X}}(x, \tilde{x})$ as follows:*

$$\epsilon \sim p_{\mathcal{E}}\,, \qquad z = s(\epsilon)\,, \qquad x = g(z)\,,$$
$$I \sim p_{\mathcal{I}}\,, \qquad \tilde{\epsilon} \sim \tilde{p}_{\tilde{\mathcal{E}}}(\tilde{\epsilon} \mid \epsilon, I)\,, \qquad \tilde{z} = \tilde{s}_I(\tilde{\epsilon})\,, \qquad \tilde{x} = g(\tilde{z})\,. \qquad (1)$$

*Here we parameterize stochastic interventions on $I$ by $\tilde{\epsilon}_i \sim p_{\mathcal{E}_i}$ for $i \in I$ and $\tilde{\epsilon}_i = \epsilon_i$ for $i \notin I$.*

**Identifiability result** The main theoretical result of this paper is that an LCM $\mathcal{M}$ can be identified from $p(x, \tilde{x})$ up to a relabeling and elementwise transformations of the causal variables:

**Theorem 1** (Identifiability of $\mathbb{R}$-valued LCMs from weak supervision). *Let $\mathcal{M} = \langle \mathcal{C}, \mathcal{X}, g, \mathcal{I}, p_{\mathcal{I}} \rangle$ and $\mathcal{M}' = \langle \mathcal{C}', \mathcal{X}, g', \mathcal{I}', p'_{\mathcal{I}'} \rangle$ be LCMs with the following properties:*

- *The LCMs have an identical observation space $\mathcal{X}$.*
- *The SCMs $\mathcal{C}$ and $\mathcal{C}'$ both consist of $n$ real-valued endogeneous causal variables and corresponding exogenous noise variables, i.e. $\mathcal{E}_i = \mathcal{Z}_i = \tilde{\mathcal{Z}}_i' = \mathcal{E}_i' = \mathbb{R}$.*
- *The intervention sets $\mathcal{I}$ and $\mathcal{I}'$ consist of all atomic, perfect interventions, $\mathcal{I} = \{\emptyset, \{z_0\}, \ldots, \{z_n\}\}$ and similar for $\mathcal{I}'$.*
- *The intervention distribution $p_{\mathcal{I}}$ and $p'_{\mathcal{I}'}$ have full support.*

*Then the following two statements are equivalent:*

1. *The LCMs entail equal weakly supervised distributions, $p_{\mathcal{M}}^{\mathcal{X}}(x, \tilde{x}) = p_{\mathcal{M}'}^{\mathcal{X}}(x, \tilde{x})$.*
2. *The LCMs are equivalent, $\mathcal{M} \sim \mathcal{M}'$.*

Let us summarize the key steps of our proof, which we provide in its entirety in Appendix A.2. The direction $2 \Rightarrow 1$ follows from the definition of equivalence. The direction $1 \Rightarrow 2$ is proven constructively along the following steps:

1. We begin by defining a diffeomorphism $\varphi = g'^{-1} \circ g : \mathcal{Z} \rightarrow \mathcal{Z}'$ and note that if $z, \tilde{z} \sim p_{\mathcal{C}}^{\mathcal{Z}}(z, \tilde{z})$, the weakly supervised distribution of causal variables of model $\mathcal{C}$, then $\varphi(z), \varphi(\tilde{z}) \sim p_{\mathcal{C}'}^{\mathcal{Z}'}(z', \tilde{z}')$. The distribution over $z, \tilde{z}$ is a mixture, where each intervention $I = \{i\}$ gives a mixture component; each component is supported on different a $(n+1)$-dimensional submanifold. Therefore there exists a bijection between the components $\psi : [n] \rightarrow [n]$ that maps intervention targets $I$ in $\mathcal{M}$ to intervention targets $I' = \psi(I)$ in $\mathcal{M}'$. Furthermore, because the joint distribution $z, \tilde{z}$ is preserved by $\varphi$, first mapping with $\varphi$, then intervening, $\mathcal{Z} \xrightarrow{\varphi} \mathcal{Z}' \xrightarrow{I'} \widetilde{\mathcal{Z}}'$, equals $\mathcal{Z} \xrightarrow{I} \widetilde{\mathcal{Z}} \xrightarrow{\varphi} \widetilde{\mathcal{Z}}'$.

2. Because $I = \{i\}$ is a hard intervention, for the order $\mathcal{Z} \xrightarrow{\varphi} \mathcal{Z}' \xrightarrow{I'} \widetilde{\mathcal{Z}}'$, $\tilde{z}'_{i'}$ is independent of $z'$. Thus in both orders, $\tilde{z}'_{i'}$ is independent of $z$. This means that for the path through $\widetilde{\mathcal{Z}}$, the intervention sample $\tilde{z}_i$ is transformed into $\tilde{z}'_{i'}$ independently of $z$. For $\mathbb{R}$-valued variables, this statistical independence implies that the transformation is constant in $z$, and thus $\varphi(z)_{i'}$ is constant in $z_j$ for $j \neq i$. $\varphi$ is therefore an elementwise reparametrization.

3. Using this, it is easy to show that $\psi$ is a causal graph isomorphism and that it is compatible with the causal mechanisms. This proves LCM equivalence $\mathcal{M} \sim \mathcal{M}'$.

---

[1]This construction is closely related to twinned SCMs (Bongers et al., 2021, Def. 2.17), typically used to compute counterfactual queries $p(\tilde{z}_{\setminus I} | z, \tilde{z}_I)$. We instead focus on the joint distribution of pre-intervention and post-intervention data.

## 3 EXPERIMENTS

The identifiability result in Sec. 2 suggests that one can identify an LCM by finding the maximum-likelihood solution of $p(x, \tilde{x})$. We now investigate whether we can indeed learn LCMs in practice. The datasets, models, and training setup are described in more detail in Appendix B.

**Relaxation of the weakly supervised setting** In realistic systems, the "before" and "after" state of a system are recorded some time apart. We do not expect that the causal variables unaffected by an intervention remain *perfectly* invariant during this time. Similarly, we want to allow for causal descendants of the intervention targets to slightly change from the exact value predicted by the weakly supervised process in Eq. (1). We model this by applying a Gaussian convolution with small variance $\sigma^2$ to $p(z, \tilde{z} \mid I)$ on all dimensions of $\tilde{z}$ that are not intervened on, resulting in a continuous density. This relaxation means that there is a gap between the requirements of our identifiability theorem is and the experimental setting. In particular, relaxing the exact manifold in the weakly supervised data space to a "fuzzy" one renders our argument for the identifiability of noise encodings and intervention targets invalid; see Brehmer & Cranmer (2020) for a related discussion. Nevertheless, we empirically find that we can still reliably identify LCMs in this more realistic setup.

Table 1: Experiment results, comparing LCMs to unstructured $\beta$-VAEs and disentanglement VAEs (dVAE), best results in bold. For Causal3DIdent, we average over six datasets with different graphs, see Appendix B. We measure disentanglement with the DCI score $D$. $D \approx 1$ implies disentanglement succeeds only with our method. LCMs learn the correct causal graph most of the time, visible as a structural Hamming distance SHD $\approx 0$. The quality of intervention inference is evaluated with the intervention negative log posterior ($-\log p_I$).

| Method | $D$ | SHD | $-\log p_I$ |
|---|---|---|---|
| *2D toy data* | | | |
| LCM | **0.99** | **0.0** | **0.28** |
| dVAE | 0.35 | n/a | 0.33 |
| $\beta$-VAE | 0.00 | n/a | n/a |
| *Causal3DIdent* | | | |
| LCM | **0.98** | **0.17** | **0.20** |
| dVAE | 0.57 | n/a | 1.98 |
| $\beta$-VAE | 0.38 | n/a | n/a |

**LCM implementation** In practice, we implement LCMs as VAEs (Kingma & Welling, 2014). In the simplest version, the causal graph $\mathcal{G}$ is fixed. The SCM and interventional distributions define the relaxed prior $p(z, \tilde{z}|I)$. Rather than with a diffeomorphism $g$, we map causal latents to observed variables through a stochastic encoder $q(z|x)$ and decoder $p(x|z)$. We then maximize the following lower bound on the weakly supervised log likelihood:

$$\log p(x, \tilde{x}) \geq \underset{\substack{z \sim q(z|x) \\ \tilde{z} \sim q(\tilde{z}|\tilde{x})}}{\mathbb{E}} \Big[ \log \underset{I \sim p_{\mathcal{I}}(I)}{\mathbb{E}} [p(z, \tilde{z}|I)] + \log p(x|z) + \log p(\tilde{x}|\tilde{z}) - \log q(z|x) - \log q(\tilde{z}|\tilde{x}) \Big]. \quad (2)$$

For our experiments with small graphs, we instantiate one such LCM per graph structure, train them on the VAE loss in Eq. (2), and select the model with the lowest validation loss. To scale to larger graphs, it will be beneficial to learn a belief over graphs and to sample adjacency matrices and intervention targets from suitable priors, for instance using Gumbel-Softmax methods (Jang et al., 2017; Brouillard et al., 2020) or other approaches (Lippe et al., 2022a); we will explore this in future work.

Following common practice in causal discovery (Zheng et al., 2018; Brouillard et al., 2020; Lippe

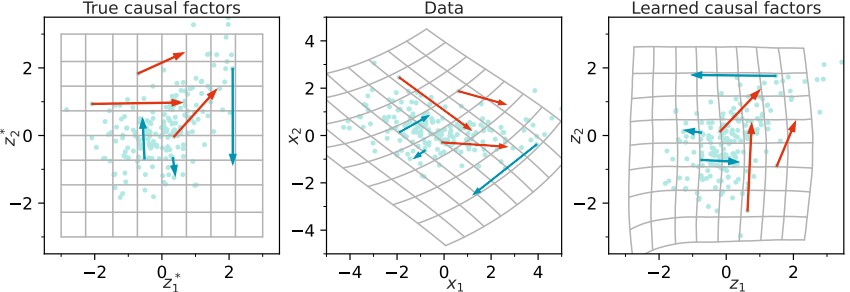

Figure 3: 2D toy data with graph $z_1^* \to z_2^*$. The grey grids show the map between true causal factors, data, and latent causal factors learned by the LCM, with graph $z_2 \to z_1$. The mint dots indicate the observational data distribution, the arrows from $z$ to $\tilde{z}$ show interventions targeting $z_1^*$ (red) or $z_2^*$ (blue). The fact that axis-aligned lines in the true latent space are mapped to axis-aligned lines in the learned latent space implies that the disentanglement succeeded.

et al., 2022a), we incentivize learning the sparsest graph compatible with the data distribution by adding a regularization term proportional to the number of edges in the graph to the loss.

**Baselines**  We compare LCMs to an unstructured $\beta$-VAE that treats $x$ and $\tilde{x}$ as i. i. d. and uses a standard Gaussian prior. We also compare to a disentanglement VAE, which models our weakly supervised process for a trivial causal graph (i. e. independent factors of variation), similar to the method proposed by Locatello et al. (2020).

**2D toy experiment**  We first test LCMs in a toy experiment with $\mathcal{X} = \mathcal{Z} = \mathbb{R}^2$. Training data is generated from a nonlinear SCM with the graph $z_1 \rightarrow z_2$ and mapped to the data space through a randomly initialized normalizing flow.

An LCM trained in the weakly supervised setting is able to reconstruct the causal factors and the causal graph accurately up to a permutation of the two variables, as shown in the Fig. 3. It fits the weakly supervised data distribution with a better log likelihood than the acausal baselines. In Tbl. 1 we quantify the quality of the learned representations with the DCI disentanglement score $D$ (Eastwood & Williams, 2018). We find that our LCM is able to disentangle the causal factors almost perfectly, while the baselines, which assume independent factors of variation, fail as expected. Finally, we test whether the learned LCM correctly infers the interventions by computing the intervention posterior $p(I|x, \tilde{x})$ and evaluating it for the true intervention $I^*$, again finding better results for the causal LCM than for the baselines.

**Causal3DIdent**  Next, we test LCMs on an adaptation of the Causal3DIdent dataset (von Kügelgen et al., 2021), which contains images of three-dimensional objects under variable positions and lighting conditions. We consider three latent variables representing object hue, the spotlight hue, and the position of the spotlight. We consider six versions of this dataset, each with a different causal graph, randomly initialized nonlinear structure functions, and heteroskedastic noise. These are mapped to images with a resolution of $64 \times 64$, see Fig. 4 for examples.

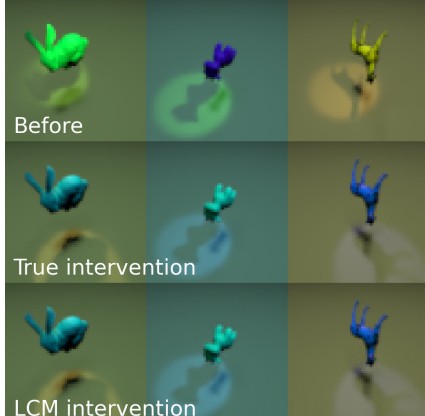

Figure 4: Causal3DIdent samples before (top row) and after (middle row) interventions, and post-intervention samples generated from the LCM under the intervention inferred from the data, indicating we correctly learned to intervene.

An LCM with convolutional encoder and decoder is again able to learn the causal variables and the causal structure between them, finding the correct causal graph in all but one settings[2]. The results in Tbl. 1 and Fig. 4 show that the learned representations are more disentangled than those learned by methods that do not account for causal structure and that the LCM can infer interventions more reliably than the baselines.

## 4 DISCUSSION

We have presented a method for causal representation learning in a weakly supervised setting, in which data consist of a system before and after a random intervention. We have proven that in this setting both the causal variables and the causal structure between them is identifiable up to permutations and elementwise reparameterizations of the causal variables. This extends the results by Locatello et al. (2020) from independent factors of variation (trivial causal graphs) to arbitrary causal structures. Our identifiability result relies on a few key assumptions, including that interventions are perfect, that all atomic interventions may be observed, and that the causal variables are real-valued. We discuss these requirements and their potential relaxation in Appendix A.3.

In practice, LCMs can be implemented in a variational autoencoder. We demonstrated in first experiments that this lets us reliably learn causal variables and nonlinear causal structure from unstructured pixel data.

---

[2]In the one configuration where our algorithm did not learn the correct graph, it instead learned a supergraph of the true graph, with one additional edge. We attribute this to insufficient regularization.

**Acknowledgments** We want to thank Dominik Neuenfeld and Frank Rösler for useful discussions and Gabriele Cesa, Yang Yang, and Yunfan Zhang for helping with our experiments.

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

# A  IDENTIFIABILITY RESULT

## A.1  DEFINITIONS

Here we define objects and relations that were not formally defined in the main body of the paper, but are necessary to make Thm. 1 precise and to prove it.

We use the following notation:

- $[n] = \{1, ..., n\}$
- $\mathbf{pa}_i^{\mathcal{C}} \subseteq [n]$ the set of parent nodes of node $i$ in graph $\mathcal{G}(\mathcal{C})$.
- $\mathbf{desc}_i^{\mathcal{C}} \subseteq [n]$ the set of descendant nodes of node $i$ in graph $\mathcal{G}(\mathcal{C})$, excluding $i$ itself.
- $\mathbf{anc}_i^{\mathcal{C}} \subseteq [n]$ the set of ancestor nodes of node $i$ in graph $\mathcal{G}(\mathcal{C})$, excluding $i$ itself.
- $\mathbf{nonanc}_i^{\mathcal{C}} = [n] \setminus (\mathbf{anc}_i^{\mathcal{C}} \cup \{i\})$ the set of non-ancestor nodes of node $i$ in graph $\mathcal{G}(\mathcal{C})$, excluding $i$ itself.
- Given measure $p$ on space $A$ and measurable function $f : A \to B$, $f_* p$ is the push-forward measure on $B$.

We describe causal structure with SCMs.

**Definition 4** (Structural causal model (SCM)). *An SCM is a tuple $\mathcal{C} = \langle \mathcal{Z}, \mathcal{E}, F, p_{\mathcal{E}} \rangle$ consisting of the following:*

- *domains $\mathcal{Z} = \mathcal{Z}_1 \times \cdots \times \mathcal{Z}_n$ of causal (endogenous) variables $z_1, \ldots, z_n$;*
- *domains $\mathcal{E} = \mathcal{E}_1 \times \cdots \times \mathcal{E}_n$ of noise (exogeneous) variables $\epsilon_1, \ldots, \epsilon_n$;*
- *a directed acyclic graph $\mathcal{G}(\mathcal{C})$, whose nodes are the causal variables and edges represent causal relations between the variables;*
- *causal mechanisms $F = \{f_1, \ldots, f_n\}$ with $f_i : \mathcal{E}_i \times \prod_{j \in \mathbf{pa}_i} \mathcal{Z}_j \to \mathcal{Z}_i$; and*
- *a probability measure $p_{\mathcal{E}}(\epsilon) = p_{\mathcal{E}_1}(\epsilon_1) \, p_{\mathcal{E}_2}(\epsilon_2) \ldots p_{\mathcal{E}_n}(\epsilon_n)$ with full support that admits a continuous density.*

*Additionally, we assume that $\forall i, \forall z_{\mathbf{pa}_i}, f_i(\cdot, z_{\mathbf{pa}_i}) : \mathcal{E}_i \to \mathcal{Z}_i$ is a diffeomorphism.*

We will need to reason about vectors being "equal up to permutation and elementwise reparameterizations". We formalize this in the following definition:

**Definition 5** ($\psi$-diagonal). *Let $\psi : [n] \to [n]$ be a bijection (that is, a permutation). Let $\varphi : \prod_{i=1}^n X_i \to \prod_{i=1}^n Y_i$ be a function between product spaces. Then $\varphi$ is $\psi$-diagonal if there exist functions, called components, $\varphi_i : X_i \to Y_{\psi(i)}$ such that $\forall i, \forall x, \varphi(x_1, ..., x_i, ..., x_n)_{\psi(i)} = \varphi_i(x_i)$.*

This lets us define isomorphisms between SCMs:

**Definition 6** (Isomorphism of SCMs). *Let $\mathcal{C} = \langle \mathcal{Z}, \mathcal{E}, F, p_{\mathcal{E}} \rangle$ and $\mathcal{C}' = \langle \mathcal{Z}', \mathcal{E}', F', p'_{\mathcal{E}} \rangle$ be SCMs. An isomorphism $\varphi : \mathcal{C} \to \mathcal{C}'$ consists of*

1. *a graph isomorphism $\psi : \mathcal{G}(\mathcal{C}) \to \mathcal{G}(\mathcal{C}')$ that tells us how to identify corresponding variables in the two models and which preserves parents: $\mathbf{pa}_{\psi(i)}^{\mathcal{C}'} = \psi(\mathbf{pa}_i^{\mathcal{C}})$ and*

2. $\psi$-*diagonal diffeomorphisms for noise and endogenous variables that tell us how to repa-rameterize them* $\varphi_{\mathcal{E}} : \mathcal{E} \to \mathcal{E}'$ *and* $\varphi_{\mathcal{Z}} : \mathcal{Z} \to \mathcal{Z}'$, *where* $\varphi_{\mathcal{E}}$ *must be measure preserving* $p_{\mathcal{E}'} = \varphi_{\mathcal{E}*} p_{\mathcal{E}}$. *For notational simplicity, we will drop the subscript in* $\varphi_{\mathcal{Z}}$ *and use the symbol* $\varphi$ *to refer both to the SCM isomorphism and the noise isomorphism.*

*The elementwise diffeomorphisms are required to make the following diagrams commute* $\forall i, i' = \psi(i)$:

$$\begin{array}{ccc} \mathcal{Z}_{\mathbf{pa}_i} \times \mathcal{E}_i & \xrightarrow{(\varphi_{\mathbf{pa}_i}, \varphi_{\mathcal{E},i})} & \mathcal{Z}'_{\mathbf{pa}_{i'}} \times \mathcal{E}'_{i'} \\ \downarrow{f_i} & & \downarrow{f'_{i'}} \\ \mathcal{Z}_i & \xrightarrow{\varphi_i} & \mathcal{Z}'_{i'} \end{array} \qquad (3)$$

*Intuitively, this says that if we apply a causal mechanism* $f_i$ *and then reparameterize the causal variable* $i$ *using* $\varphi_i$, *we get the same thing as first reparameterizing the parents and noise variable of variable* $i$, *and then applying the causal mechanism* $f'_{i'}$.

To reason about interventions, we equip SCMs with intervention distributions in the following definition.

**Definition 7** (Intervention structural causal model (ISCM)). *An intervention structural causal model (ISCM) is a tuple* $\mathcal{D} = \langle \mathcal{C}, \mathcal{I}, p_{\mathcal{I}} \rangle$ *of*

1. *an acyclic SCM* $\mathcal{C} = \langle \mathcal{Z}, \mathcal{E}, F, p_{\mathcal{E}} \rangle$ *that admits a faithful distribution, meaning that conditional independence of causal variables* $z$ *implies* $d$-*separation (Pearl, 2000).*
2. *a set* $\mathcal{I}$ *of interventions on* $\mathcal{C}$, *where each intervention* $(I, (\tilde{f}_i)_{i \in I}) \in \mathcal{I}$ *consist of*
   (a) *a subset* $I \subset \{1, ..., n\}$ *of the causal variables, called the intervention target set, and*
   (b) *for each* $i \in I$, *a new causal mechanism* $\tilde{f}_i : \mathcal{E}_i \to \mathcal{Z}_i$ *which replaces the original mechanism and which does not depend on the parents.*
   *We define intervention set* $\mathcal{I}$ *to be atomic if the number of targeted variables is one or zero.*
3. *a probability measure* $p_{\mathcal{I}}$ *over* $\mathcal{I}$.

We can extend the notion of isomorphism from SCMs to ISCMs.

**Definition 8** (Isomorphism of ISCMs). *Let* $\mathcal{D} = \langle \mathcal{C}, \mathcal{I}, p_{\mathcal{I}} \rangle$ *and* $\mathcal{D}' = \langle \mathcal{C}', \mathcal{I}', p'_{\mathcal{I}'} \rangle$ *be ISCMs. An ISCM isomorphism is an SCM isomorphism* $\varphi : \mathcal{C} \to \mathcal{C}'$ *with underlying graph isomorphism* $\psi : \mathcal{G}(\mathcal{C}) \to \mathcal{G}(\mathcal{C}')$ *and a* $\psi$-*diagonal diffeomorphism* $\tilde{\varphi}_{\mathcal{E}} : \mathcal{E} \to \mathcal{E}$ *such that*

• *the graph isomorphism* $\psi$ *induces a bijection of intervention sets*

$$\psi_{\mathcal{I}} : \mathcal{I} \to \mathcal{I}' : (I, (\tilde{f}_i)_{i \in I}) \mapsto (\psi(I), (\tilde{f}'_{i'})_{i' \in \psi(I)})$$

• *for each intervention* $(I, (\tilde{f}_i)_{i \in I}) \in \mathcal{I}$, *and each intervened on variable* $i \in I$, *the following diagram commutes:*

$$\begin{array}{ccc} \mathcal{E}_i & \xrightarrow{\tilde{\varphi}_{\mathcal{E},i}} & \mathcal{E}'_{\psi(i)} \\ \downarrow{\tilde{f}_i} & & \downarrow{\tilde{f}'_{\psi(i)}} \\ \mathcal{Z}_i & \xrightarrow{\varphi_i} & \mathcal{Z}'_{\psi(i)} \end{array} \qquad (4)$$

• $\tilde{\varphi}_{\mathcal{E}}$ *is measure preserving, i. e.* $p_{\mathcal{E}'} = (\tilde{\varphi}_{\mathcal{E}})_* p_{\mathcal{E}}$.
• *the bijection* $\psi_{\mathcal{I}} : \mathcal{I} \to \mathcal{I}'$ *preserves the distribution over interventions:* $\psi_* p_{\mathcal{I}} = p'_{\mathcal{I}'}$.

Latent Causal Models (LCMs), defined in Def. 1, add a map to the data space to an ILCM. We can lift ISCM isomorphisms to LCM isomorphisms by requiring that these decoders must respect the ISCM isomorphism.

**Definition 9** (Isomorphism of LCMs). *Let* $\mathcal{M} = \langle \mathcal{C}, \mathcal{X}, g, \mathcal{I}, p_{\mathcal{I}} \rangle$ *and* $\mathcal{M}' = \langle \mathcal{C}', \mathcal{X}, g', \mathcal{I}', p'_{\mathcal{I}'} \rangle$ *be LCMs with identical observation space* $\mathcal{X} = \mathcal{X}'$. *An LCM isomorphism of LCM is an ISCM isomorphism* $\varphi : \mathcal{D} \to \mathcal{D}'$ *such that the decoders respect the SCM isomorphism, so this diagram must commute:*

$$\begin{array}{ccc} \mathcal{Z} & \xrightarrow{\varphi} & \mathcal{Z}' \\ & {}_{g} \searrow \quad \swarrow {}_{g'} & \\ & \mathcal{X} & \end{array} \qquad (5)$$

$$\begin{array}{ccc}
\epsilon, \tilde{\epsilon} & \xrightarrow{\varphi_{\mathcal{E}}, \tilde{\varphi}_{\mathcal{E}}} & \epsilon', \tilde{\epsilon}' \\
{\scriptstyle s, \tilde{s}_I} \downarrow & & \downarrow {\scriptstyle s', \tilde{s}'_{I'}} \\
z, \tilde{z} & \dashrightarrow{\varphi_{\mathcal{Z}}, \varphi_{\mathcal{Z}}} & z, \tilde{z}' \\
{\scriptstyle g, g} \downarrow & & \downarrow {\scriptstyle g', g'} \\
x, \tilde{x} & =\!\!=\!\!=\!\!= & x, \tilde{x}
\end{array}$$

Figure 5: An illustration of the spaces and maps in our definitions and proof. When LCMs $\mathcal{M}, \mathcal{M}'$ are isomorphic, all squares in the diagram should commute. Additionally, all maps should preserve the weakly supervised distributions on the variables and all horizontal maps should be $\psi$-diagonal. Note that the latent variables $(\epsilon, z)$ can differ up to a diffeomorphism, but the $x$ variables are actually observed, so must be identically equal. From that equality, the other horizontal maps are uniquely defined.

*Remark* 1. By defining objects and isomorphisms, we have defined a groupoid of SCMs, a groupoid of ISCMs and a groupoid of LCMs, as the isomorphisms are composed and inverted in an obvious way.

**Definition 10** (Equivalence). *We call two SCMs, ISCMs, or LCMs equivalent if an isomorphism exists between them.*

Informally, two SCMs, ISCMs, or LCMs are equivalent if there is a $\psi$-diagonal map between their causal variables (i. e. the causal variables are equal up to permutation and elementwise diffeomorphisms), there is a $\psi$-diagonal map between their noise encodings, and all other structure (decoders, intervention sets, intervention distributions) is compatible with these reparameterizations.

Next, we define the solution function of an SCM or ISCM, which maps from noise variables to causal variables by repeatedly applying the causal mechanisms.

**Definition 11** (Solution). *Given an ISCM $\mathcal{D} = \langle \mathcal{C}, \mathcal{I}, p_{\mathcal{I}} \rangle$, the solution function $s : \mathcal{E} \to \mathcal{Z}$ is the unique function such that for all $i \in [n]$, the following diagram commutes (Bongers et al., 2021)*

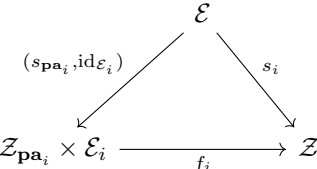

*Or equations, we have that $s(\epsilon)_i = f(\epsilon_i; s(\epsilon_{\mathbf{pa}_i}))$. Similarly, intervention $(I, (\tilde{f}_i)_{i \in I}) \in \mathcal{I}$ yields a solution function $\tilde{s}_I : \mathcal{E} \to \mathcal{Z}$ with the modified causal mechanisms.*

For example, with two variables with $z_1 \to z_2$, the solution is given by:

$$s : \mathcal{E} \to \mathcal{Z} : \begin{pmatrix} \epsilon_1 \\ \epsilon_2 \end{pmatrix} \mapsto \begin{pmatrix} z_1 \\ z_2 \end{pmatrix} = \begin{pmatrix} f_1(\epsilon_1) \\ f_2(\epsilon_2, f_1(\epsilon_1)) \end{pmatrix}.$$

Since we require causal mechanisms to be pointwise diffeomorphic, the solution function is a diffeomorphism as well.

Pushing the noise distribution of an SCM through the solution function finally gives us the (observable) distribution entailed by an SCM or ISCM. In an ISCM or LCM we can define several other (observable or interventional) distributions.

**Definition 12** (Distributions). *Given an LCM $\mathcal{M} = \langle \mathcal{C}, \mathcal{X}, g, \mathcal{I}, p_{\mathcal{I}} \rangle$, we have the following generative process:*

$$\epsilon \sim p_{\mathcal{E}}, \qquad z = s(\epsilon), \qquad x = g(z), \qquad e = s^{-1}(z)$$
$$I \sim p_{\mathcal{I}}, \qquad \tilde{\epsilon} \sim \tilde{p}_{\tilde{\mathcal{E}}}(\tilde{\epsilon} \mid \epsilon, I), \qquad \tilde{z} = \tilde{s}_I(\tilde{\epsilon}), \qquad \tilde{x} = g(\tilde{z}), \qquad \tilde{e} = s^{-1}(\tilde{z}). \qquad (6)$$

*where $p(\tilde{\epsilon}_i \mid \epsilon_i, i \in I) = p_{\mathcal{E}_i}(\tilde{\epsilon}_i)$ and $p(\tilde{\epsilon}_i \mid \epsilon_i, i \notin I) = \delta(\tilde{\epsilon}_i \mid \epsilon_i)$ is the Dirac measure.*

*Then we define the following weakly supervised distributions:*

- *The weakly supervised noise distribution with interventions: $p_{\mathcal{C}}^{\mathcal{E},\mathcal{I}}(\epsilon, \tilde{\epsilon}, I)$.*
- *The weakly supervised causal distribution with interventions: $p_{\mathcal{C}}^{\mathcal{Z},\mathcal{I}}(z, \tilde{z}, I)$.*
- *The weakly supervised observational distribution with interventions: $p_{\mathcal{M}}^{\mathcal{X},\mathcal{I}}(x, \tilde{x}, I)$.*

*These distributions are given by appropriate pushforwards of the noise distributions through the transformations in Eq. (6).*

*By marginalizing over I, we get $p_{\mathcal{C}}^{\mathcal{E}}, p_{\mathcal{C}}^{\mathcal{Z}}, p_{\mathcal{C}}^{e}, p_{\mathcal{M}}^{\mathcal{X}}$ respectively.*

The relationships between all the maps can be found in Fig. 5.

## A.2  IDENTIFIABILITY PROOF

First, we prove two auxiliary lemmata.

**Lemma 1.** *Let $f : [0, 1] \to [0, 1]$ be differentiable and Lebesgue measure preserving. Then either $f(x) = x$ or $f(x) = 1 - x$.*

*Proof.* We follow the proof from Stack Exchange user zhw (2016). Let $\lambda$ be the Lebesgue measure. Measure preservation means that for any measurable subset $U \subseteq [0, 1]$, $\lambda(U) = \lambda(f^{-1}(U))$.

First, note that $f$ is surjective, because otherwise the image of $f$ is a proper subinterval $[a, b] \subsetneq [0, 1]$ and $\lambda(f^{-1}([a, b])) = \lambda([0, 1]) = 1 > \lambda([a, b]) = b - a$, which contradicts measure-preservation.

Define the open ball $B(x, r) = \{y \in [0, 1] \mid |y - x| < r\}$. Suppose that $f'(0) = 0$ for some $x \in [0, 1]$. Then there exists an $r > 0$ such that $f(B(x, r)) \subseteq B(f(x), r/4)$, and thus $B(x, r) \subseteq f^{-1}(B(f(x), r/4))$. Therefore, $r \leq \lambda(B(x, r)) \leq \lambda(f^{-1}(B(f(x), r/4)))$, while $\lambda(B(f(x), r/4)) \leq 2 \cdot r/4 = r/2$, contradicting measure preservation. Hence $f'(x) \neq 0$ on $[0, 1]$.

By the Darboux theorem, $f'$ is either strictly positive or strictly negative on the interval and thus $f$ is either strictly increasing or decreasing and thus a bijection. Assume that it is strictly increasing, then $\forall x \in [0, 1], x = \lambda([0, x]) = \lambda(f^{-1}(f([0, x]))) = \lambda(f([0, x])) = f(x) - f(0) = f(x)$. Similarly, if it is strictly decreasing, we find $f(x) = 1 - x$. $\qquad\square$

**Lemma 2.** *Let $A = C = \mathbb{R}$ and $B = \mathbb{R}^n$. Let $f : A \times B \to C$ be differentiable. Define differentiable measures $p_A$ on $A$ and $p_C$ on $C$. Let $\forall b \in B, f(\cdot, b) : A \to C$ be measure-preserving. Then $f$ is constant in $B$.*

*Proof.* Let $P_A : A \to [0, 1], P_C : C \to [0, 1]$ be the diffeomorphic cumulative density functions. Then $P_A^{-1}$ and $P_C^{-1}$ are measure-preserving maps from the uniform distribution on $[0, 1]$. Now write $g : [0, 1] \times B \to [0, 1] : (z, b) \mapsto P_C(f(P_A^{-1}(z), b))$ such that this diagram of measure-preserving differentiable maps commutes:

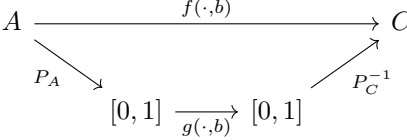

Then $g$ is differentiable and $\forall b \in B$ measure-preserving $[0, 1] \to [0, 1]$. By the previous Lemma 1, the only differentiable measure-preserving functions $[0, 1] \to [0, 1]$ are $\mathrm{id}$ and $1 - \mathrm{id}$. As $g$ is continuous in $B$, it can not vary between $\mathrm{id}$ and $1 - \mathrm{id}$ and thus $g$, and consequently $f$ are constant in $B$. $\qquad\square$

We can interpret this lemma in terms of statistical independence. Starting from a product measure on $A \times B$, the requirements of the lemma correspond to $a \perp\!\!\!\perp b$ and $c \perp\!\!\!\perp b$. The lemma thus defines a sense in which for real-valued variables, statistical independence implies functional independence (the converse is always true).

Now in the remainder of this subsection, we prove the main theorem.

**Theorem 1** (Identifiability of $\mathbb{R}$-valued LCMs from weak supervision). *Let $\mathcal{M} = \langle \mathcal{C}, \mathcal{X}, g, \mathcal{I}, p_{\mathcal{I}} \rangle$ and $\mathcal{M}' = \langle \mathcal{C}', \mathcal{X}, g', \mathcal{I}', p'_{\mathcal{I}'} \rangle$ be LCMs with the following properties:*

- *The SCMs $\mathcal{C}$ and $\mathcal{C}'$ both consist of $n$ real-valued endogenous variables, i. e. $\mathcal{E}_i = \mathcal{Z}_i = \mathcal{Z}_i' = \mathcal{E}_i' = \mathbb{R}$.*
- *The intervention sets $\mathcal{I}$ and $\mathcal{I}'$ consist of the empty intervention and all atomic interventions, $\mathcal{I} = \{\emptyset, \{z_0\}, \dots, \{z_n\}\}$ and similar for $\mathcal{I}'$.*
- *The intervention distribution $p_{\mathcal{I}}$ and $p_{\mathcal{I}'}'$ have full support.*

*Then the following two statements are equivalent:*

1. *The weakly supervised distributions entailed by the LCMs are equal, $p_{\mathcal{M}}(x, \tilde{x}) = p_{\mathcal{M}'}(x, \tilde{x})$.*
2. *The LCMs are equivalent, $\mathcal{M} \sim \mathcal{M}'$.*

*Proof.* "$(2) \Rightarrow (1)$": If the LCMs are equivalent, then the fact that $\varphi_{\mathcal{E}}$ and $\tilde{\varphi}_{\mathcal{E}}$ are measure preserving and that diagrams (3) and (4) commute, implies that $p_{\mathcal{C}'}^{\mathcal{Z}'} = (\varphi_{\mathcal{Z}}, \varphi_{\mathcal{Z}})_* p_{\mathcal{C}}^{\mathcal{Z}}$. Then because diagram (5) commutes, the weakly supervised distributions coincide, $p_{\mathcal{M}'}^{\mathcal{X}} = p_{\mathcal{M}}^{\mathcal{X}}$.

"$(1) \Rightarrow (2)$": Conversely, if the weakly supervised distributions coincide, $p_{\mathcal{M}'}^{\mathcal{X}} = p_{\mathcal{M}}^{\mathcal{X}}$, the images of $g : \mathcal{Z} \to \mathcal{X}, g' : \mathcal{Z}' \to \mathcal{X}$ coincide,

$$\varphi = g'^{-1} \circ g : \mathcal{Z} \to \mathcal{Z}' \tag{7}$$

is a diffeomorphism, and $\phi$ preserves the weakly supervised distribution over causal variables: $p_{\mathcal{C}'}^{\mathcal{Z}'} = (\varphi, \varphi)_* p_{\mathcal{C}}^{\mathcal{Z}}$.

LCM equivalence then follows from showing that $\varphi : \mathcal{D} \to \mathcal{D}'$ is an ISCM isomorphism, where $\mathcal{D} = \langle \mathcal{C}, \mathcal{I}, p_{\mathcal{I}} \rangle$ and $\mathcal{D}' = \langle \mathcal{C}', \mathcal{I}', p_{\mathcal{I}'}' \rangle$ be the ISCMs inherent to $\mathcal{M}$ and $\mathcal{M}'$. We show this in the following steps:

1. For each intervention $I$ in $\mathcal{D}$, there is a corresponding intervention $I'$ in $\mathcal{D}'$, given by a permutation $\psi : [n] \to [n]$, such that $\varphi$ preserves the interventional distribution.
2. The diffeomorphism $\varphi$ is $\psi$-diagonal.
3. The permutation $\psi$ preserved the ancestry structure of graphs $\mathcal{G}(\mathcal{C})$ and $\mathcal{G}(\mathcal{C}')$.
4. The diffeomorphism $\varphi_{\mathcal{E}} : \mathcal{E} \to \mathcal{E}$ of noise variables is $\psi$-diagonal.
5. The causal mechanisms are compatible with $\varphi$.

**Step 1: Interventions preserved** Remember that the diffeomorphism $\varphi : \mathcal{Z} \to \mathcal{Z}'$ is such that $p_{\mathcal{C}'}^{\mathcal{Z}'} = (\varphi, \varphi)_* p_{\mathcal{C}}^{\mathcal{Z}}$. For atomic interventions $I \neq J \in \mathcal{I}$, consider the intersection of the supports of the weakly supervised distribution for interventions on $I$ and $J$: $U = \operatorname{supp} p_{\mathcal{C}}^{\mathcal{Z}, \mathcal{I}}(z, \tilde{z} \mid I) \cap \operatorname{supp} p_{\mathcal{C}}^{\mathcal{Z}, \mathcal{I}}(z, \tilde{z} \mid J) \subset \mathcal{Z} \times \mathcal{Z}$. Note that $U$ has zero measure in $p_{\mathcal{C}}^{\mathcal{Z}, \mathcal{I}}(U \mid I) = p_{\mathcal{C}}^{\mathcal{Z}, \mathcal{I}}(U \mid J) = 0$. The distribution is thus a discrete mixture on $(z, \tilde{z})$ of non-overlapping distributions.

The diffeormorphism $(\varphi, \varphi)$ must map between these mixtures. Thus there exists a bijection $\psi : \mathcal{I} \to \mathcal{I}'$, also inducing a permutation $\psi : [n] \to [n]$, such that

$$p_{\mathcal{C}'}^{\mathcal{Z}', \mathcal{I}'} = (\varphi, \varphi, \psi)_* p_{\mathcal{C}}^{\mathcal{Z}, \mathcal{I}} \ .$$

**Step 2: $\varphi$ is $\psi$-diagonal** This measure preservation lets us define two equal distributions on $\mathcal{Z} \times \tilde{\mathcal{Z}}' \times \mathcal{I}$, namely $(\operatorname{id}_{\mathcal{Z}}, \varphi, \operatorname{id}_{\mathcal{I}})_* p_{\mathcal{C}}^{\mathcal{Z}, \mathcal{I}}$ and $(\varphi^{-1}, \operatorname{id}_{\tilde{\mathcal{Z}}'}, \psi^{-1})_* p_{\mathcal{C}'}^{\mathcal{Z}', \mathcal{I}'}$. In particular, these must then have equal conditionals $p(\tilde{z}' \mid z, I)$. Thus, for any $U \subseteq \tilde{\mathcal{Z}}', z \in \mathcal{Z}, I \in \mathcal{I}$,

$$p_{\mathcal{C}'}^{\mathcal{Z}', \mathcal{I}'}(\tilde{z}' \in U \mid \varphi(z), \psi(I)) = p_{\mathcal{C}}^{\mathcal{Z}, \mathcal{I}}(\tilde{z} \in \varphi^{-1}(U) \mid z, I)$$

The conditional probability $p_{\mathcal{C}}^{\mathcal{Z}, \mathcal{I}}(\tilde{z} \mid z, I)$ can be interpreted as a stochastic map $\mathcal{Z} \to \tilde{\mathcal{Z}}$. The above relation can then be written as a commuting diagram of stochastic maps, $\forall I \in \mathcal{I}, I' = \psi(I)$:

$$
\begin{array}{ccc}
\mathcal{Z} & \xrightarrow{p_{\mathcal{C}}^{\mathcal{Z}, \mathcal{I}}(\tilde{z} \mid z, I)} & \tilde{\mathcal{Z}} \\
\downarrow{\scriptstyle \varphi} & & \downarrow{\scriptstyle \varphi} \\
\mathcal{Z}' & \xrightarrow[p_{\mathcal{C}'}^{\mathcal{Z}', \mathcal{I}'}(\tilde{z}' \mid z', I')]{} & \tilde{\mathcal{Z}}'
\end{array}
\tag{8}
$$

where we treat $\varphi : \mathcal{Z} \to \mathcal{Z}'$ as a deterministic stochastic map.

For any variable $i \in [n]$, write the other nodes as $o = [n] \setminus \{i\}$. Let $I = \{i\}$. Then $p_{\mathcal{C}}^{\mathcal{Z},\mathcal{I}}(\tilde{z} \mid z, I)$ can be written as a string diagram of stochastic maps:

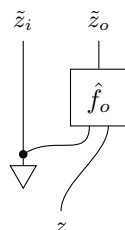

This string diagram represents a conditional probability distribution $p(\tilde{z}_i, \tilde{z}_o \mid z)$ and is read from the bottom to the top. String diagrams map formally to a generative process (Fritz, 2020) and have been used previously in the context of causal models (Fong, 2013). In this case, the diagram maps to:

$$\tilde{z}_i \sim p(\tilde{z}_i), \quad \tilde{z}_o = \hat{f}_o(\tilde{z}_i, z)$$

where $p(\tilde{z}_i)$ is the interventional distribution and the deterministic map $\hat{f}_o : \widetilde{\mathcal{Z}}_i \times \mathcal{Z} \to \widetilde{\mathcal{Z}}_o$ can be constructed from the inverse solution $s^{-1} : \mathcal{Z} \to \mathcal{E}$ and the causal mechanisms. Each box in a string diagram of stochastic maps denotes a stochastic map and each line to a measurable space. The triangle is the stochastic map $\star \to \widetilde{\mathcal{Z}}_i$ (the star denoting the one-point space; maps from which correspond to probability distributions over the codomain). The $\bullet$ represents copying a variable.

The above commuting diagram (8) can then be written as the equality of the following two string diagrams, where $\psi(I) = I' = \{i'\}, o' = [n] \setminus \{i'\}$. We write $\varphi : \mathcal{Z} \to \mathcal{Z}'$ as the pair $\varphi_{i'} : \mathcal{Z} \to \mathcal{Z}'_{i'}, \varphi_{o'} : \mathcal{Z} \to \mathcal{Z}'_{o'}$ obtained by projecting the output of $\varphi$ to the partition $\mathcal{Z}' = \mathcal{Z}'_{i'} \times \mathcal{Z}'_{o'}$:

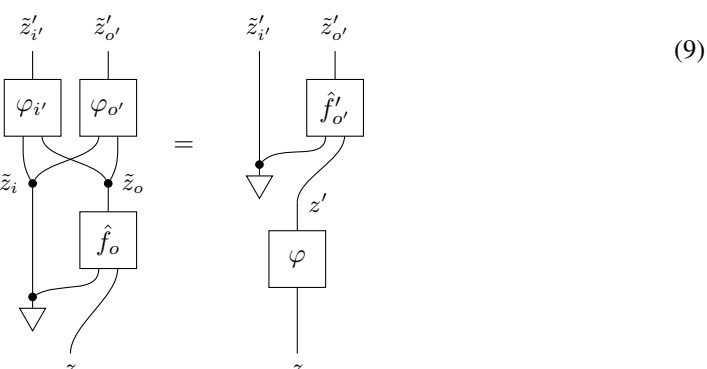

$$(9)$$

This should be read as the equality of the two conditional probability distributions $p(\tilde{z}'_{i'}, \tilde{z}'_{o'} \mid z)$ generated in the following way:

$$\text{Left:} \quad \tilde{z}_i \sim p(\tilde{z}_i), \quad \tilde{z}_o = \hat{f}_o(\tilde{z}_i, z), \quad \tilde{z}'_{i'} = \varphi(\tilde{z}_i, \tilde{z}_o)_{i'}, \quad \tilde{z}_{o'} = \varphi(\tilde{z}_i, \tilde{z}_o)_{o'}.$$
$$\text{Right:} \quad z' = \varphi(z), \quad \tilde{z}'_{i'} \sim p'(\tilde{z}'_{i'}), \quad \tilde{z}'_{o'} = f'_{o'}(\tilde{z}'_{i'}, z').$$

The string diagram equality (9) implies equality when we disregard outputs $\mathcal{Z}'_{o'}$:

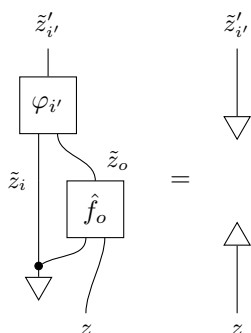

where the upwards pointing triangle represents discarding a variable.

Using Lemma 2, and the fact that $\widetilde{\mathcal{Z}}_i = \widetilde{\mathcal{Z}}'_{i'} = \mathbb{R}$, the composed differentiable function $\widetilde{\mathcal{Z}}_i \times \mathcal{Z} \to \widetilde{\mathcal{Z}}'_{i'}$ is constant in $\mathcal{Z}$. Thus we have a deterministic function $\widetilde{\mathcal{Z}}_i \to \widetilde{\mathcal{Z}}'_{i'}$ such that:

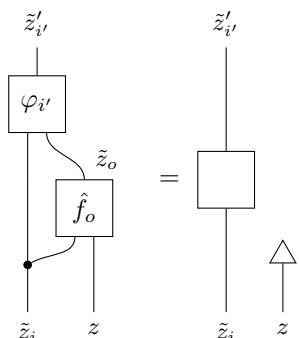

The deterministic function $\widetilde{\mathcal{Z}}_i \times \mathcal{Z} \to \widetilde{\mathcal{Z}}_i \times \widetilde{\mathcal{Z}}_o$ is surjective and both the left- and right-hand side can be seen as first applying this function (though the output is discarded on the right hand side), which implies there exists a function $\widetilde{\mathcal{Z}}_i \to \widetilde{\mathcal{Z}}'_{i'}$ such that

$$
\begin{array}{ccc}
\tilde{z}'_{i'} & & \tilde{z}'_{i'} \\
\boxed{\varphi_{i'}} & = & \boxed{\phantom{xx}} \quad \triangle \\
\tilde{z}_i \quad \tilde{z}_o & & \tilde{z}_i \quad \tilde{z}_o
\end{array}
$$

In words, the function $\varphi_{i'} : \mathcal{Z}_i \times \mathcal{Z}_o \to \mathcal{Z}'_{i'}$ is constant in $\mathcal{Z}_o$. This holds for all $i$ and thus $\varphi$ is $\psi$-diagonal.

**Step 3: Ancestry preserved**   Let $i \neq j \in [n]$, $i' = \psi(i)$, $j' = \psi(j)$, and $I = \{i\}$. Writing $\varphi$ as $\psi$-diagonal, the commuting diagram (8) for the $j'$ component of $\tilde{z}'$, can be written as the following string diagram:

$$
\begin{array}{ccc}
\tilde{z}'_{j'} & & \tilde{z}'_{j'} \\
\boxed{\varphi_{j'}} & & \boxed{\hat{f}'_{j'}} \\
\tilde{z}_j & = & \tilde{z}'_{i'} \quad z' \\
\boxed{\hat{f}_j} & & \boxed{\varphi} \\
\tilde{z}_i \quad z & & z
\end{array}
$$

The left hand side is a deterministic map $\mathcal{Z} \to \widetilde{\mathcal{Z}}'_{j'}$ if and only if $\hat{f}_j$ is constant in $\widetilde{\mathcal{Z}}_i$ which by faithfulness is the case if and only if $i \notin \mathbf{anc}_j$. The same holds on the right hand side, so $\forall i \neq j \in [n]$, $i \in \mathbf{anc}^{\mathcal{C}}_j \iff \psi(i) \in \mathbf{anc}^{\mathcal{C}'}_{\psi(j)}$.

**Step 4: Noise map diagonal**   Define $\varphi_{\mathcal{E}} = s'^{-1} \circ \varphi \circ s : \mathcal{E} \to \mathcal{E}'$. Note that $\varphi_{\mathcal{E}}(\epsilon)_{i'}$ only depends on $\epsilon_i$ and $\epsilon_{\mathbf{anc}_i}$, because $s(\epsilon)_{\mathbf{anc}_i, i}$ and $s'^{-1}(z')_{i'}$ only depend on ancestors, $\varphi$ is $\psi$-diagonal and $\psi$ preserves ancestry.

The map $\varphi$ is measure-preserving. Thus $\forall i$ and writing $A = \mathbf{anc}_i$, the conditional $p(z_i \mid z_A) = p(z_i \mid z_{\mathbf{pa}_i})$, interpreted as a stochastic map, is preserved by $\varphi$. We can express this as another

commuting diagram, in which the two paths from $\mathcal{E}_A$ to $\mathcal{E}'_{i'}$ must be equal:

$$\mathcal{E}_A \xrightarrow{s_A} \mathcal{Z}_A \xrightarrow{p(z_i|z_{\mathbf{pa}_i})} \mathcal{Z}_{A,i}$$

$$\varphi_A \downarrow \qquad\qquad \downarrow \varphi_{A,i}$$

$$\mathcal{Z}'_{A'} \xrightarrow{p(z'_{i'}|z'_{\mathbf{pa}_{i'}})} \mathcal{Z}'_{A',i'} \xrightarrow{f'_{i'}{}^{-1}} \mathcal{E}'_{i'}$$

where $f'_{i'}{}^{-1}(z') = f(z'_{\mathbf{pa}_{i'}}, \cdot)^{-1}(z'_{i'})$. Then we have:

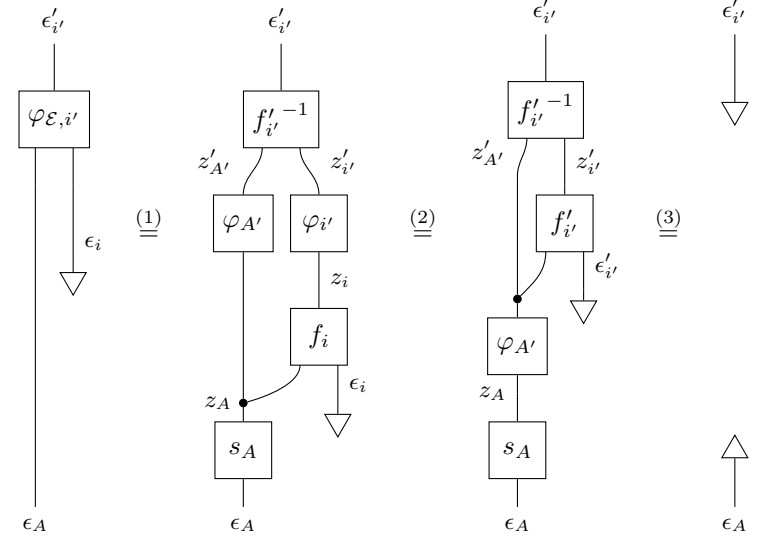

where the first equality follows from the definition of $\varphi_{\mathcal{E},i'}$, the second equality from the commuting diagram above and the third equality from the fact that $f'_{i'}$ and $f'_{i'}{}^{-1}$ cancel. Then, again using Lemma 2, the map on the left hand side must be constant in $\epsilon_A$. The noise encoding is thus also $\psi$-diagonal.

**Step 5: Equivalence**  Consider for a variable $i$ and with $i' = \psi(i)$ the following commuting diagram of deterministic maps. Note that we write the causal mechanism $f_i$ as a function of all ancestors, not just the parents, so it is constant in the non-parents. Because of faithfulness, it is non-constant in the parents. Since $\psi$ preserves ancestors, $f'_{i'}$ is well-typed.

$$
\begin{array}{ccc}
\mathcal{E} & \xrightarrow{\varphi_{\mathcal{E}}} & \mathcal{E}' \\
\downarrow{(s_{\mathbf{anc}_i}, \mathrm{id}_{\mathcal{E}_i})} & & \downarrow{(s'_{\mathbf{anc}_{i'}}, \mathrm{id}_{\mathcal{E}'_{i'}})} \\
\mathcal{Z}_{\mathbf{anc}_i} \times \mathcal{E}_i & \xrightarrow{(\varphi_{\mathcal{Z},\mathbf{anc}_i}, \varphi_{\mathcal{E},i})} & \mathcal{Z}'_{\mathbf{anc}_{i'}} \times \mathcal{E}'_{i'} \\
\downarrow{f_i} & & \downarrow{f'_{i'}} \\
\mathcal{Z}_i & \xrightarrow{\varphi_{\mathcal{Z},i}} & \mathcal{Z}'_{i'}
\end{array}
$$

The composition of the left vertical maps is equal to $s_i$, the composition of the right vertical maps to $s'_{i'}$. Therefore and because of the definition of $\varphi_{\mathcal{E}}$, the outer and the top square commute. Then, because $(s_{\mathbf{anc}_i}, \mathrm{id}_{\mathcal{E}_i})$ is surjective, the bottom square also commutes (Riehl, 2017, Lemma 1.6.21).

Then for $z_j \in \mathbf{anc}_i$, we have that

$$z_j \in \mathbf{pa}_i^{\mathcal{C}} \iff f_i \text{ not constant in } z_j \iff f'_{i'} \text{ not constant in } z'_{j'} \iff z'_{j'} \in \mathbf{pa}_{i'}^{\mathcal{C}'}.$$

And thus $\psi$ not only preserves ancestry, but also parenthood and is thus a graph isomorphism $\psi : \mathcal{G}(\mathcal{C}) \to \mathcal{G}(\mathcal{C}')$. Diagram (3) commutes, and we have established an SCM isomorphism $\varphi : \mathcal{C} \to \mathcal{C}'$.

To have this also be an ISCM isomorphism, we need diagram (4) to commute and the distribution

over interventions to be preserved. For the first, use the fact that all maps in (4) are isomorphisms to simply define $\tilde{\varphi}_{\mathcal{E}}$ so that the diagram commutes. The second follows directly from the assumptions. Hence $\varphi : \mathcal{D} \to \mathcal{D}'$ is an ISCM isomorphism, $\mathcal{D} \sim \mathcal{D}'$, and—together with the arguments in the beginning of this proof—finally $\mathcal{M} \sim \mathcal{M}'$. $\qquad\square$

### A.3 LIMITATIONS & GENERALIZATION

Our identifiability result relies on a few assumptions. Here we discuss some key requirements of Thm. 1 and whether they can be relaxed.

**Diffeomorphic causal mechanisms** In Def. 4, we require causal mechanisms to be pointwise diffeomorphisms from noise variables to causal variables. Under some mild smoothness assumptions, any SCM can be brought into this form by elementwise redefinitions of the variables, without affecting the observational or interventional distributions. (However, such a redefinition may change counterfactual distributions.)

**Perfect interventions** Our proof of Thm. 1 requires perfect interventions, i.e. intervened-upon mechanisms not depending on any causal variables. This is arguably the biggest mismatch between our asumptions and many real-world systems.

**Diffeomorphic decoder** Definition 1 and Thm. 1 assume that the map from causal variables to observed data is given by a deterministic, diffeomorphic decoder. However, our practical implementation in a VAE uses a stochastic decoder and allows for noisy data. Our experiments provide empirical evidence for identifiability in this setting. We believe that it may be possible to extend Thm. 1 to stochastic decoders, similarly to Khemakhem et al. (2020). We plan to study this extension in future work.

**Real-valued causal variables** Theorem 1 assumes real-valued causal and noise variables, $\mathcal{Z}_i = \mathcal{E}_i = \mathbb{R}$. We can easily extend this to intervals $(a, b) \in \mathbb{R}$, as these are isomorphic to $\mathbb{R}$. However, the extension to arbitrary continuous spaces or $\mathbb{R}^n$ is not straightforward. The main reason is that our proof relies on Lemma 2, which does not generalize.

Let us provide a counterexample for identifiability with circle $S^1$-valued causal variables.

**Example 1** ($S^1$-valued non-identifiable LCMs)**.** *Consider an LCM $\mathcal{M} = \langle \mathcal{C}, \mathcal{X}, g, \mathcal{I}, p_{\mathcal{I}} \rangle$ with the following components:*

- *The SCM $\mathcal{C}$ consists of two circle-valued variables $z_1, z_2 \in S^1$ with noise variables $\epsilon_1, \epsilon_2 \in S^1$. We parameterize $S^1$ as $[0, 2\pi)$ with addition defined modulo $2\pi$.*
- *The causal graph is $z_1 \to z_2$.*
- *The causal mechanisms are $f_1(\epsilon_1) = \epsilon_1$ and $f_2(\epsilon_2; z_1) = \epsilon_2 + z_1$.*
- *The solution function is $s(\epsilon_1, \epsilon_2) = (\epsilon_1, \epsilon_2 + \epsilon_1)$.*
- *The noise variables are distributed as $\epsilon_1 \sim \mathcal{U}$, uniformly, and $\epsilon_2 \sim q$, which we require to not be invariant under translations (so in particular not uniform). For example, one can take the von Mises distribution $\log q(\epsilon_2) = \cos(\epsilon_2) + \text{const}$.*
- *The observation space is $\mathcal{X}$ and the decoder $g : S^1 \times S^1 \to \mathcal{X}$ is diffeomorphic.*
- *The intervention set $\mathcal{I}$ consists of the empty intervention, atomic interventions on $z_1$ with $\tilde{z}_1 \sim \mathcal{U}$, and atomic interventions on $z_2$ with $\tilde{z}_2 \sim \mathcal{U}$. Each of these interventions has probability $\frac{1}{3}$ in $p_{\mathcal{I}}$.*

*Note that the SCM is faithful, as $z_1 \not\perp\!\!\!\perp z_2$ in the observational distribution, because $q$ is not translationally invariant. The LCM entails the weakly supervised causal distribution*

$$p_{\mathcal{C}}^{\mathcal{Z}}(z, \tilde{z}) = \mathcal{U}(z_1)\, q(z_2 - z_1) \left[ \frac{1}{3}\, \delta(\tilde{z}_1 - z_1)\, \delta(\tilde{z}_2 - z_2) \right.$$

$$\left. + \frac{1}{3}\, \mathcal{U}(\tilde{z}_1)\, \delta(\tilde{z}_2 - z_2 - \tilde{z}_1 + z_1) + \frac{1}{3}\, \delta(\tilde{z}_1 - z_1)\, \mathcal{U}(\tilde{z}_2) \right] \quad (10)$$

*with Dirac delta $\delta$. The weakly supervised data distribution is then given by $p_{\mathcal{M}}^{\mathcal{X}} = (g_*, g_*) p_{\mathcal{C}}^{\mathcal{Z}}$.*

*Now consider a second LCM $\mathcal{M}' = \langle \mathcal{C}', \mathcal{X}, g', \mathcal{I}', p'_{\mathcal{I}'} \rangle$:*

- *The SCM $\mathcal{C}'$ consists of two circle-valued variables $z_1', z_2' \in S^1$ with noise variables $\epsilon_1', \epsilon_2' \in S^1$.*
- *The causal graph is trivial and the causal mechanisms are given by the identity, $f_i'(\epsilon_i') = \epsilon_i'$.*
- *The noise variables are distributed as $\epsilon_1' \sim \mathcal{U}$ and $\epsilon_2' \sim q$.*
- *The observation space is $\mathcal{X}$ and the decoder $g' : S^1 \times S^1 \to \mathcal{X}$ is given by the diffeomorphism $g'(z') = g \circ s(z')$, where $s$ is the solution function of $\mathcal{C}$.*
- *The intervention set $\mathcal{I}'$ consists of empty interventions, atomic interventions on $z_1'$ with $\tilde{z}_1' \sim \mathcal{U}$, and atomic interventions on $z_2'$ with $\tilde{z}_2' \sim \mathcal{U}$. Each of these interventions has probability $\frac{1}{3}$ in $p_{\mathcal{I}'}$.*

*We find a weakly supervised causal distribution*

$$p_{\mathcal{C}'}^{\mathcal{Z}'}(z', \tilde{z}') = \mathcal{U}(z_1') \, q(z_2') \left[ \frac{1}{3} \, \delta(\tilde{z}_1' - z_1') \, \delta(\tilde{z}_2' - z_2') \right.$$
$$\left. + \frac{1}{3} \, \mathcal{U}(\tilde{z}_1') \, \delta(\tilde{z}_2' - z_2') + \frac{1}{3} \, \delta(\tilde{z}_1' - z_1') \, \mathcal{U}(\tilde{z}_2') \right]. \quad (11)$$

*Clearly, two LCMs are not equivalent, because their graphs are non-isomorphic. Yet, if we define*

$$\varphi : \mathcal{Z} \to \mathcal{Z}' : (z_1, z_2) \mapsto (z_1, z_2 - z_1)$$

*then the weakly supervised distribution of the causal variables is preserved:*

$$((\varphi, \varphi)_* p_{\mathcal{C}}^{\mathcal{Z}})(z', \tilde{z}') = p_{\mathcal{C}}^{\mathcal{Z}}((z_1', z_2' + z_1'), (\tilde{z}_1', \tilde{z}_2' + \tilde{z}_1'))$$

$$= \mathcal{U}(z_1') \, q(z_2' + z_1' - z_1') \left[ \frac{1}{3} \, \delta(\tilde{z}_1' - z_1') \, \delta(\tilde{z}_2' + \tilde{z}_1' - (z_2' + z_1')) \right.$$

$$\left. + \frac{1}{3} \, \mathcal{U}(\tilde{z}_1') \, \delta(\tilde{z}_2' + \tilde{z}_1' - (z_2' + z_1') - \tilde{z}_1' + z_1') + \frac{1}{3} \, \delta(\tilde{z}_1' - z_1') \, \mathcal{U}(\tilde{z}_2' + \tilde{z}_1') \right]$$

$$= \mathcal{U}(z_1') \, q(z_2') \left[ \frac{1}{3} \, \delta(\tilde{z}_1' - z_1') \, \delta(\tilde{z}_2' - z_2') \right.$$

$$\left. + \frac{1}{3} \, \mathcal{U}(\tilde{z}_1') \, \delta(\tilde{z}_2' - z_2') + \frac{1}{3} \, \delta(\tilde{z}_1' - z_1') \, \mathcal{U}(\tilde{z}_2') \right]$$

$$= p_{\mathcal{C}'}^{\mathcal{Z}'}(z', \tilde{z}')$$

*where we use that the density $\mathcal{U}$ is constant. Also, because $\varphi = s^{-1}$ and $g'(z') = g \circ s(z')$, we have that $p_{\mathcal{M}}^{\mathcal{X}} = p_{\mathcal{M}'}^{\mathcal{X}}$.*

*So these two models with their non-isomorphic graph structures have identical weakly-supervised distributions on the observables $x, \tilde{x}$. They therefore provide a counter-example for a straightforward generalization of Thm. 1 to causal variables with arbitrary continuous domains.*

Why does identifiability fail in this example? It is because the interventional distributions in $\mathcal{M}$ have an accidental symmetry not expected by the graph structure, namely translational invariance. This makes it possible to fit the weakly supervised distribution with a simpler causal graph. This is related to faithfulness, but the standard definition of faithfulness only concerns observational distributions (and in this sense both $\mathcal{M}$ and $\mathcal{M}'$ are faithful). Because of this accidental symmetry, steps 1 and 3 of our proof do not hold any more.

We have circumvented such issues in Thm. 1 by requiring that all causal and noise variables are $\mathbb{R}$-valued. In this setting, functional dependence implies statistical dependence, as formalized in Lemma 2, and the counterexample does not work. We conjecture that it is possible to generalize Thm. 1 to arbitrary continuous domains under mild additional assumptions, but leave this for future work.

Finally, we believe that such accidental symmetries are unlikely in the sense that under an appropriate measure over LCMs, non-identifiable LCMs have have zero measure. We find it likely that this issue will not occur frequently in practical systems (unless these are finetuned to exhibit exactly this behaviour). Overall, we conjecture that identifiability from weak supervision can be generalized beyond the real-valued case presented in Thm. 1.

Table 2: Experiment results. For each experiment, we show the true causal graph underlying the data-generating process. We then show the results from our LCMs and compare to unstructured $\beta$-VAE and disentanglement VAE (dVAE) baselines. We show the learned causal graph, the structural Hamming distance SHD between the learned and the true graph, the DCI disentanglement score ($D$), and the intervention negative log posterior ($-\log p_I$). Best results in bold.

| Dataset | True graph | Method | Learned graph | SHD | $D$ | $-\log p_I$ |
|---|---|---|---|---|---|---|
| 2D toy data | (graph) | LCM | (graph) | **0** | **0.99** | **0.28** |
| | | dVAE | – | n/a | 0.35 | 0.33 |
| | | $\beta$-VAE | – | n/a | 0.00 | n/a |
| Causal3DIdent | (graph) | LCM | (graph) | **0** | **1.00** | **0.16** |
| | | dVAE | – | n/a | **1.00** | **0.16** |
| | | $\beta$-VAE | – | n/a | 0.55 | n/a |
| | (graph) | LCM | (graph) | **0** | **0.99** | 0.23 |
| | | dVAE | – | n/a | 0.84 | **0.22** |
| | | $\beta$-VAE | – | n/a | 0.68 | n/a |
| | (graph) | LCM | (graph) | 1 | **0.96** | **0.19** |
| | | dVAE | – | n/a | 0.20 | 2.88 |
| | | $\beta$-VAE | – | n/a | 0.06 | n/a |
| | (graph) | LCM | (graph) | **0** | **0.96** | **0.21** |
| | | dVAE | – | n/a | 0.46 | 4.31 |
| | | $\beta$-VAE | – | n/a | 0.44 | n/a |
| | (graph) | LCM | (graph) | **0** | **0.98** | **0.21** |
| | | dVAE | – | n/a | 0.62 | 0.24 |
| | | $\beta$-VAE | – | n/a | 0.35 | n/a |
| | (graph) | LCM | (graph) | **0** | **0.96** | **0.18** |
| | | dVAE | – | n/a | 0.32 | 4.06 |
| | | $\beta$-VAE | – | n/a | 0.24 | n/a |
| | Average | LCM | | **0.17** | **0.98** | **0.20** |
| | | dVAE | | n/a | 0.57 | 1.98 |
| | | $\beta$-VAE | | n/a | 0.38 | n/a |

## B  EXPERIMENT DETAILS

### B.1  2D TOY EXPERIMENT

In our first experiment, we generate latent data in $\mathcal{Z} = \mathbb{R}^2$ from a nonlinear SCM with graph $z_1 \to z_2$. In particular, we have that $z_1 \sim \mathcal{N}(z_1; 0, 1^2)$ and $z_2 \sim \mathcal{N}(z_1; 0.3z_1^2 + 0.6z_1, 0.8^2)$. This latent data is mapped through the data space with a randomly initialized coupling flow with five affine coupling layers interspersed with random permutations of the dimensions. For the weakly supervised setting we use a uniform intervention prior over $\{\emptyset, \{z_1\}, \{z_2\}\}$. We generate $10^5$ training samples, $10^5$ additional training samples for the models used to compute the DCI metrics, $10^4$ validation samples, and $10^4$ evaluation samples.

The learned LCM consists of an SCM prior, an encoder, and a decoder. In the SCM, the graph is fixed (we "learn" the graph by training multiple LCMs with different fixed graphs and then selecting the model with the best validation loss). Each causal mechanism is implemented as an MLP of the parents that outputs the parameters of an affine transformation from a standard normal noise variable to a causal variable. The encoder and decoder are diagonal Gaussians, with mean and standard deviations output by an MLP. For each MLP, we use two hidden layers with 100 units each and ReLU activations.

The disentanglement VAE baseline uses the same setup, except with a trivial graph. The $\beta$-VAE uses

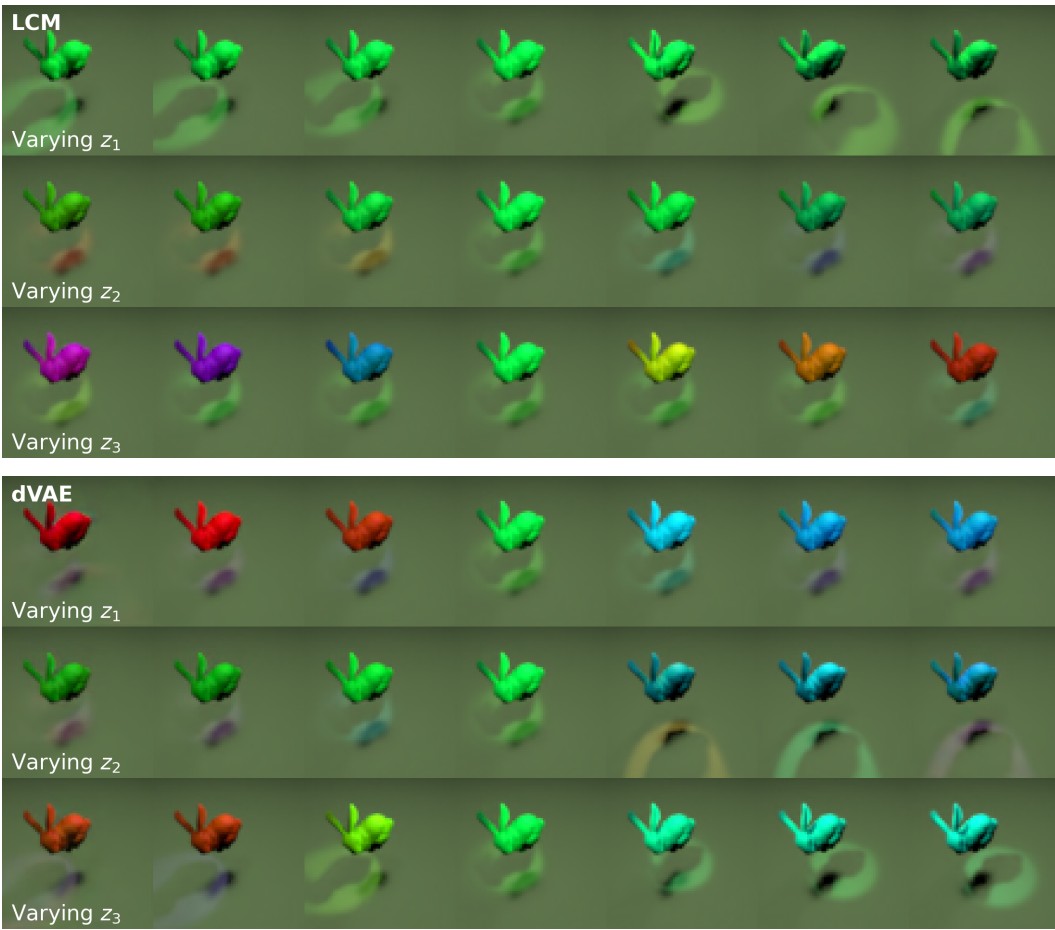

Figure 6: Effect of varying the learned causal factors on the image in the "chain" subset of the Causal3DIdent dataset. We encode a single test images (middle column) into the three learned latents, vary each of these causal factors independently, and show the reconstructed images. The LCM (top) learns a representation that is quite disentangled: $z_1$ corresponds to the spotlight position, $z_2$ to the spotlight hue, and $z_3$ to the object hue. In contrast, the acausal dVAE baseline strongly entangles these factors in its learned representation.

the same encoder and decoder, but uses a standard normal prior rather than an SCM and treats $x$ and $\tilde{x}$ as two i. i. d. samples from the same distribution.

All models are trained on the VAE loss in Eq. (2) plus a regularization term (0.1 times the number of edges in the graph). We train for $10^5$ steps using the Adam optimizer Kingma & Ba (2015) with a batch size of 100. The learning rate is initially $10^{-3}$ and is annealed with a cosine schedule. We estimate the model log likelihood using importance weighting with importance sampling (a la IWAE Burda et al. (2016)), using 10 samples at validation time and 100 samples at test time. We use a manifold "fuzziness" of $\sigma = 0.1$. For each method and each graph, we train three models with different random seeds and select the model with the best validation log likelihood.

### B.2  CAUSAL3DIDENT

In the Causal3DIdent experiments we consider six different datasets, each generated from a different causal graph, SCM, and decoder. The six causal graphs we consider are:

- the trivial graph ⚬ ⚬ ⚬,
- single edge ⚬→⚬ ⚬.

- the chain ,
- the fork ,
- the collider , and
- the full graph .

For each of these subsets, we randomly generate a nonlinear SCM with heteroskedastic noise: for each causal mechanism, we randomly initialize an MLP that outputs the scale and shift of an affine transformation as a function of the causal parents. We choose an MLP initialization scheme that emphasizes nontrivial, nonlinear causal effects. We then identify a random permutation of the three causal variables with three high-level concepts in the Causal3DIdent dataset: the object hue, the spotlight hue, and the spotlight position. We use the following causal graphs:

- single edge: object hue → spotlight position;
- chain: spotlight position → spotlight hue → object hue;
- fork: spotlight hue → spotlight position, object hue;
- collider: spotlight hue → object hue ← spotlight position;
- full graph: spotlight hue → object hue → spotlight position, spotlight hue → spotlight position.

Since all of these properties are defined on a range $[0, 2\pi)$, we apply an elementwise $\mathrm{arctanh}$ transform and rescaling to our variables such that they populate a subset of $[0, 2\pi)$. This also avoids topological issues. Next, we generate images in $64 \times 64$ resolution following the procedure described in von Kügelgen et al. (2021). We use Blender (Blender Online Community, 2021) to generate 3D rendered images based on the previously defined causal variables. To increase diversity of the six datasets, we render each dataset with a different object: Teapot (Newell, 1975), Armadillo (Krishnamurthy & Levoy, 1996), Hare (Turk & Levoy, 1994), Cow (Crane, 2021), Dragon (Curless & Levoy, 1996), and Horse (Praun et al., 2000). We generate $10^5$ training samples, $10^4$ validation samples, and $10^4$ evaluation samples.

The learned LCMs consist again of an SCM prior, which is the same as in the 2D toy experiment, an encoder, and a decoder. For the encoder and decoder we use a convolutional architecture with four residual blocks, using downsampling via average-pooling and bilinear upsampling, respectively. We do not use BatchNorm, as we found that that can lead to practical issues when images in a batch are very similar.

Our training setup is as in the 2D toy experiment, except that we use a batch size of 64, train for $2.3 \cdot 10^5$ steps, and use an initial learning rate of $3 \cdot 10^{-4}$. We find it beneficial to begin training with a lower weight of the KL divergence in the VAE loss, $\beta = 0.01$, and increasing this until the final value of $\beta = 0.1$ during the first half of training. We initialize the manifold "fuzziness" parameter $\sigma$ to 0.2 and anneal it to 0.01 over the first half of training. For each method and each graph, we train three models with different random seeds and select the model with the best validation log likelihood.

We report our results in Tbl. 2 and visualize the disentanglement properties of the learned representations in Fig. 6.

