# OpenReview forum: "Weakly supervised causal representation learning"
_ICLR.cc/2022/Workshop/OSC — ICLR2022 OSC  Poster_

### Official Review · Reviewer_oy5R · 2022-03-02
**Review. Briefly: How Far Could We Relax The Strict Assumptions For Identifiability?**

**Rating:** 2
**Confidence:** 2

**Review:**

**1. Summary and Contributions**

The paper first considers identifiability from data theoretically, then actual identification in practice. The authors refer to identifiability as their interest in recovering the structural causal model alongside a decoding model (which maps latent factors to the original data space) when being given mixed (meaning both observational and interventional) data from the original data space. The paper proves said identifiability under strict assumptions. The paper further provides an empirical investigation in support of the identifiability theorem using toy and semi-realistic data sets to corroborate the identifiability result.

**2. Strengths**

The paper has several noteworthy strengths, considered one-by-one in the following list (the list is ordered in correspondence to the paper presentation):

* The proposed problem is of high interest, since progress in learning causal representations is an arguably crucial step for developing next generation learning systems.
* The motivation is written clearly as the authors reference works that are relevant (even determining) for their proposed ideas.
* The proposed identifiability result, which suggests that optimizing the given mixed data is sufficient for recovering the necessary causal quantities of interest, since it provides both certainty for settings in which any practitioner might satisfy the provided assumptions plus the techniques used for proving this key theorem might be used for future relaxations of the theorem.
* Corroborating the identifiability result with practical empirics on synthetic data (sanity check) but also semi-realistic data which involves actual low-level pixel observations of images that are high-dimensional and carry aleatoric uncertainty.
* The critical discussion of the key assumptions in the appendix, since assumptions like perfect, atomic interventions or observing all interventions are arguably very strict and an average practitioner should not be expecting these assumptions to hold for general problem settings of interest.

**3. Weaknesses**

The paper suffers from several disadvantages (ranging in importance from minor to more fundamental) that however IMHO can be improved upon mostly quickly as they are mostly aspects of presentation. Thereby, the following list - again one-by-one - aims to provide specific pointers with improvement suggestions if applicable (please note, the list is unordered):

* The official OSC workshop submission guideline suggested an unlimited appendix officially but asked for the authors to only include minor details. This setting does not apply here since the authors provide the proof of the theorem in its entirety within the appendix where said theorem is the first key and arguably main result of the paper. Furthermore, 12 pages of appendix over 5 pages of main paper seems disproportionate to what I would believe the workshop organizers intended with the original statement on limitations for the submissions. Nonetheless, I've considered the complete appendix for my review, but have noted for myself to be aware of potential biases or unfairness when reviewing other papers for OSC that do not violate the given limitations and recommendations. IMHO both the story line of this work and their presentation lend themselves better to an extended treatise that might go beyond the format of common conference papers and rather into that of common journals.
* The authors might consider saving presentation space for this OSC workshop by avoiding a definition like the SCM in its entirety as done in Def.1 on p.2 or overview-/introductory-type of paragraphs to their subsections when they are identifiable from the given context (e.g. first paragraph in Sec.2). The extra space can then be rescheduled for e.g. including key elements of the actual proof which otherwise had to be resorted to the appendix.
* The authors might consider aligning their notation with common notation used in causality literature like Pearl 2009, Peters et al. 2017 or Bareinboim et al. 2020 to foster consistency within the community, since there does not seem to be a particular advantageous reason for choosing otherwise.
* The mostly informally stated definition of the LCM isomorphism, which would benefit from either more rigor or aided schematic visualization.
* The key assumptions are arguably very strict and therefore restrict the relevance of the certainty provided by the proven theorem.

**3. Correctness, Clarity, and Literature**

No contradictions or any sort of relevant mistake have been detected in the paper. Existing bodies of work are being referenced accordingly.

**4. Reproducibility, Code Release, and Assumptions**

Sufficient details for reproduction are being provided. Unfortunately, without actual code. All key assumptions for the method are being pointed out explicitly (in the appendix).

---

### Official Review · Reviewer_Cuc6 · 2022-03-18

**Rating:** 2
**Confidence:** 2

**Review:**

The paper focuses on an interesting problem of learning causal models from high-dimensional unstructured data with interventions, the authors provided indentifiability proofs for such a setting with weak supervision signals (only the pair of data before and after intervention is given, the intervention itself can be unknown). Here are my comments for the paper.

1. The paper focuses on an very interesting and important problem. The authors provided identifiability proofs for such a setting with only weak supervision signals, the intervention itself can be unknown, this is an very interesting result.

2. However, given such strong and interesting theoretical proofs, the experiment seems to be a bit pale in comparison. The experiment is only conducted on a 2D toy setting, and my main concern is that the causal graph in the experiment is fixed and not learned, and learning the causal graph structure is one of the most important (and challenging) aspect of causal learning.

I understand that this might be prelimary work, however, I do feel that this should be described properly in the introduction and abstract. From the way it currently reads, it seems as if the authors have both theoretical and experimental results indicating that a SCM can be fully learned from high-dimensional unstructured data. This is a little misleading. It would be nice if the authors could update the paper to reflect the experimental results.

That being said, I think the theoretical results are intriguing. I would be excited to see further results, especially experimental results showing that a SCM can be learned from high dimensional data.

---

### Decision · Program_Chairs · 2022-03-21

**Decision:**

Accept (Poster)

**Comment:**

The reviewers agree the paper should be accepted at the workshop. Congratulations!